# Do exercise-associated genes explain phenotypic variance in the three components of fitness? a systematic review & meta-analysis

Henry C. Chung[1]*, Don R. Keiller[2], Justin D. Roberts[1], Dan A. Gordon[1]

1 Cambridge Centre for Sport & Exercise Sciences, Anglia Ruskin University, Cambridge, United Kingdom,
2 School of Life Sciences, Anglia Ruskin University, Cambridge, United Kingdom

* henry.chung@pgr.anglia.ac.uk

## Abstract

The aim of this systematic review and meta-analysis was to identify a list of common, candidate genes associated with the three components of fitness, specifically cardiovascular fitness, muscular strength, and anaerobic power, and how these genes are associated with exercise response phenotype variability, in previously untrained participants. A total of 3,969 potentially relevant papers were identified and processed for inclusion. After eligibility and study selection assessment, 24 studies were selected for meta-analysis, comprising a total of 3,012 participants (male n = 1,512; females n = 1,239; not stated n = 261; age 28 ± 9 years). Meta-Essentials spreadsheet 1.4 (Microsoft Excel) was used in creating the forest plots and meta-analysis. IBM SPSS statistics V24 was implemented for the statistical analyses and the alpha was set at p ≤ 0.05. 13 candidate genes and their associated alleles were identified, which were associated with the phenotypes of interest. Analysis of training group data showed significant differential phenotypic responses. Subgroup analysis showed; 44%, 72% and 10% of the response variance in aerobic, strength and power phenotypes, respectively, were explained by genetic influences. This analysis established that genetic variability explained a significant proportion of the adaptation differences across the three components of fitness in the participants post-training. The results also showed the importance of analysing and reporting specific gene alleles. Information obtained from these findings has the potential to inform and influence future exercise-related genes and training studies.

## Introduction

Current evidence shows that cardiovascular fitness, muscular strength, and anaerobic power are key in determining an individuals' health-related fitness, well-being, and quality of life, as well as successful performance in many sporting events [1–5]. For example, the $\dot{V}O_{2max}$, is a key index of cardiovascular and cardiorespiratory fitness and an increase in $\dot{V}O_{2max}$, improves

**Data Availability Statement:** All relevant data are within the manuscript and its Supporting Information files.

**Funding:** The author(s) received no specific funding for this work.

**Competing interests:** The authors have declared that no competing interests exist.

the ability of the body to both supply and utilise oxygen. This prolongs the time to exhaustion and the ability to sustain aerobic exercise for longer periods of time, at higher intensities [6]. For some, this could mean the difference between being able to walk upstairs easily, rather than with effort and associated discomfort. Below average, $\dot{V}O_{2max}$ is also correlated to increased morbidity and mortality and improvement in $\dot{V}O_{2max}$ can prevent early mortality [7]. There is a similar rationale for improving both strength and power as well. Strength refers to the force that can be generated during a voluntary muscle contraction and is required for everyday tasks and mobility [2,5]. Anaerobic power is the ability of the neuromuscular system to produce the greatest possible action in a given time period and is needed for quick bursts explosive movements and agility [8]. Hence, it is beneficial for individuals to improve these components, irrespective of their initial level of fitness and especially, for those classed as untrained [4–6,9–12].

Although it has always been a key objective of health and exercise sciences to improve these specific key components of fitness, Schutte et al. [13] and Sarzynski, Ghosh and Bouchard, [14] stress that an individuals' responsiveness to exercise training varies significantly, depending on the precise exercise-stimuli given. In this connection, previous studies show that a genetic component, in the response to exercise training, can explain up-to 80% of the variability in aerobic, strength and power adaptations [2,15–17]. Such findings suggest that the current health and exercise guidelines, promoting generic fitness classes and group exercises, are of questionable value, without consideration of individuals' genetic profile. Accordingly, several well-studied genes, show significant associations with exercise trainability and successful increases in performance, providing advantages in sports and athletic competitions [18,19]. Many other genes have been shown to influence all aspects of fitness, including, but not limited to, energy-pathways, metabolism, storage, cell growth, protein, hormonal, and enzyme interactions [13,19–21]. All such genes have been termed 'candidate genes' [14,22–25] and may be useful indicators in predicting and producing successful training responses, to a particular exercise intervention and maximising health benefits. However, the difficulty is identifying and selecting key genes from the extensive suite of candidate genes, shown to be associated with exercise responses [13,15,17,26]. Since most current research typically relies on studies that only investigate a limited number of genes and/or single genes and their alleles, and mainly in twins or elite and high-level performance, a literature review may well be more suited in identifying multiple genes and their alleles and their applications on the untrained [17,27,28]. Thus, Ahmetov et al. [29] and Williams and Folland, [30] reported that, regardless of the high heritability of exercise response, no single gene, or polymorphism, has been shown to be solely responsible for a particular physiological variable, due to the large number of genetic polymorphisms associated with aerobic, strength and power phenotypes.

Accordingly, the aim of this review is to identify candidate genes and their alleles that best define exercise phenotype responses to training interventions, with respect to the three components of fitness, in the untrained population.

## Methods

This review was conducted in accordance with the Cochrane guidelines of systematic reviews and the Preferred Reporting Items for Systematic Reviews and Meta-Analyses (PRISMA) [31–33]. This review was ethically approved by the Faculty of Science and Engineering Research Ethics Panel at Anglia Ruskin University, Cambridge, UK. A comprehensive literature search was performed using Scopus, Web of Science, PubMed and SportDiscus, between June and July 2020. Multiple database thesauruses and MeSH terms were employed to pool keywords for the systematic search. An email alarm system, on each database, was created for

notifications on new publications related to the search terms. All relevant studies were exported and sorted within a bibliographic management system (Refworks, ProQuest, USA) and an automatic deduplication tool was applied. Studies' inclusion was screened in accordance with title, abstract and full text. Potential studies that met the inclusion criteria were further inspected [34]. The reference list of studies and grey literature was also explored, any missing information was requested from study authors, *via* email, if possible, otherwise results could not be included.

## Eligibility criteria

The following PICOS criteria (Population/Participant, Intervention, Comparison, Outcome and Study type) [35] were implemented. Firstly, a minimum effective intervention time-course of two weeks (six sessions) was identified from studies such as, Astorino and Schubert, [26] and Hautala et al. [2]. Relevant outcomes were defined as the common phenotype measurements across all studies [3,5,17].

Population: (a) health untrained, human male and female participants with no stated medical conditions; (b) aged between 18–55 years; (c) training status below the ACSM norm values for cycle ergometer cardio-respiratory fitness ($\dot{V}O_{2max}$), one repetition maximum (1RM) leg-press and cycle ergometer peak power output (PPO), ($\dot{V}O_{2max} \leq 45$ ml·kg$^{-1}$·min$^{-1}$ for Males; $\leq$ 40 for females. Lower body 1RM $\leq$ 1.91 x mass for males; $\leq$ 1.32 x mass for females. PPO $\leq$ 9.22 W/kg for males; $\leq$ 7.65 W/kg for females) [5].

Intervention: (d) training $\geq$ two weeks (minimum six sessions, three per-week); (e) include either: 1) continuous endurance/aerobic Interval training, 2) resistance, or weight training, or 3) interval/sprint/anaerobic training; (f) no dietary manipulation.

Comparison: (g) Pre *vs* Post changes in primary phenotypes; (h) participants grouped by gene, or genotype.

Outcome: (i) change in primary variable(s): $\dot{V}O_{2max}$, lower body 1RM or PPO.

Study type: (j) repeated measures method; (k) original research study; (l) English language. Studies that did not meet all the above PICOS criteria were excluded from this review.

## Quality and risk of bias assessment

A COnsensus-based Standards for the selection of health status Measurement INstruments (COSMIN) checklist was implemented to evaluate the transparency and risk of bias of the studies, by measuring methodological quality [36]. The COSMIN 'worst score' approach was set for all items at $\geq$ 3, to meet the acceptable requirement of study quality and study selection [31]. To confirm the consistency and reliability of the COSMIN tool, two reviewers (HC and DG) independently evaluated the studies for inclusion. Each COSMIN item for all categories was scored from 4–1 (31 items total), where 4 was low and 1 was high risk, respectively, and any disagreements from authors were resolved using the mean COSMIN scores. Studies were only included if $\geq$3. 95% Limits of Agreement (LoA) were calculated using the Bland and Altman approximate method [37]. At least two studies, or groups, were required to report the same gene of interest to establish a conclusive outcome measurement, otherwise this could not be evaluated, by meta-analysis [38,39].

## Statistical analysis

Means, Standard Deviations (SD), Standard Errors (SE) and 95% Confidence Intervals (CI) were extracted from all studies and pooled for analysis. Pre-to-post intervention scores for each study were converted into Effect Sizes (ES) and Standardised Mean Differences

(SMD), to express intervention effects and genotypic variance between groups in standard-ised units. Pooled standard deviation, SE, variance ($s^1$), 95% Cl and the weighted Mean Cohen's d, were also calculated: classifying d values as 0.20–0.49 small, 0.50–0.79 medium and $\geq$ 0.80 large effect [39–41]. Meta-Essentials spreadsheet 1.4 (Microsoft Excel 2016, Washington, USA) was used for the meta-analysis and creation of forest plots. IBM SPSS statistics version 24 (SPSS, Chicago, Illinois) was used for all other statistical testing, with alpha set at p $\leq$ 0.05. For the assessment of any genetic effects, a subgroup stratification analysis was implemented. Here the training groups in each study were combined and then split, based on genotypes, and then further analysed and compared. Normality and homoge-neity of variance were calculated *via* Shapiro-Wilk test and Levene's test, respectively. Where necessary, a non-parametric Kruskal-Wallis H test was used to determine if there were any significant differences between the gene groups. Partial Eta Squared and Mean Ranks were used to determine the variability within subgroup genetics and the estimation of gene and allelic variability and their contribution towards the change in training phenotypes.

## Results

A comprehensive flow diagram representing the study retrieval process and exclusions was created (Fig 1). The figure also outlines the process of the PICOS and the COSMIN checklist between reviewers. 29 studies were initially included for meta-analysis, however a further five studies were excluded, as the genes in that study were only reported on one occasion [42–46]. The final 24 studies contained 89 groups (43 aerobic; 29 strength; 17 power), with a total of 3,012 participants, and 13 candidate genes and associated alleles.

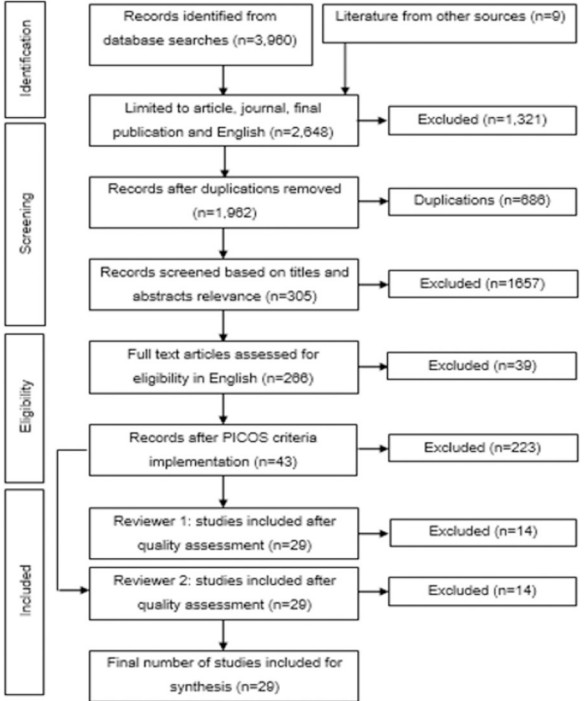

**Fig 1. Flow diagram.** Step-by-step method of collecting and excluding studies at each stage for this review. Also, where other sources from unpublished material and grey literature were entered. This entire process was repeated twice.

## Genes associated with cardiorespiratory fitness ($\dot{V}O_{2max}$)

$\dot{V}O_{2max}$ increased by 10.97 ± 3.80%, with aerobic training interventions, across all studies in this review. Forest plot analysis demonstrated that, irrespective of the genes, the overall results represented a medium to large effect-size change in $\dot{V}O_{2max}$, which was classed as a very highly significant improvement with training intervention alone (p < 0.001) (Fig 2). On average, these studies [12,18,24,25,47–56] used durations of 36 minutes, with intensities of 74% $\dot{V}O_{2max}$, or 77% HRmax, performed over 3-days a week and 12 weeks of training.

Between subgroup analysis revealed non-normal distributions across the nine aerobic associated gene groups: D(43), .876, $p < 0.05$. Here, Kruskal-Wallis H testing found significant differences between the gene groups, H(8), 18.427, $p = 0.018$. Partial Eta Squared, calculated by Pearson's $\chi 2$ tests, found that 44% of the total variability in the increase in $\dot{V}O_{2max}$ post-training intervention was explained by nine gene subgroups (Table 1). Post-Hoc subgroup analysis shows the distribution of the variance in $\dot{V}O_{2max}$ scores between the genetic groups.

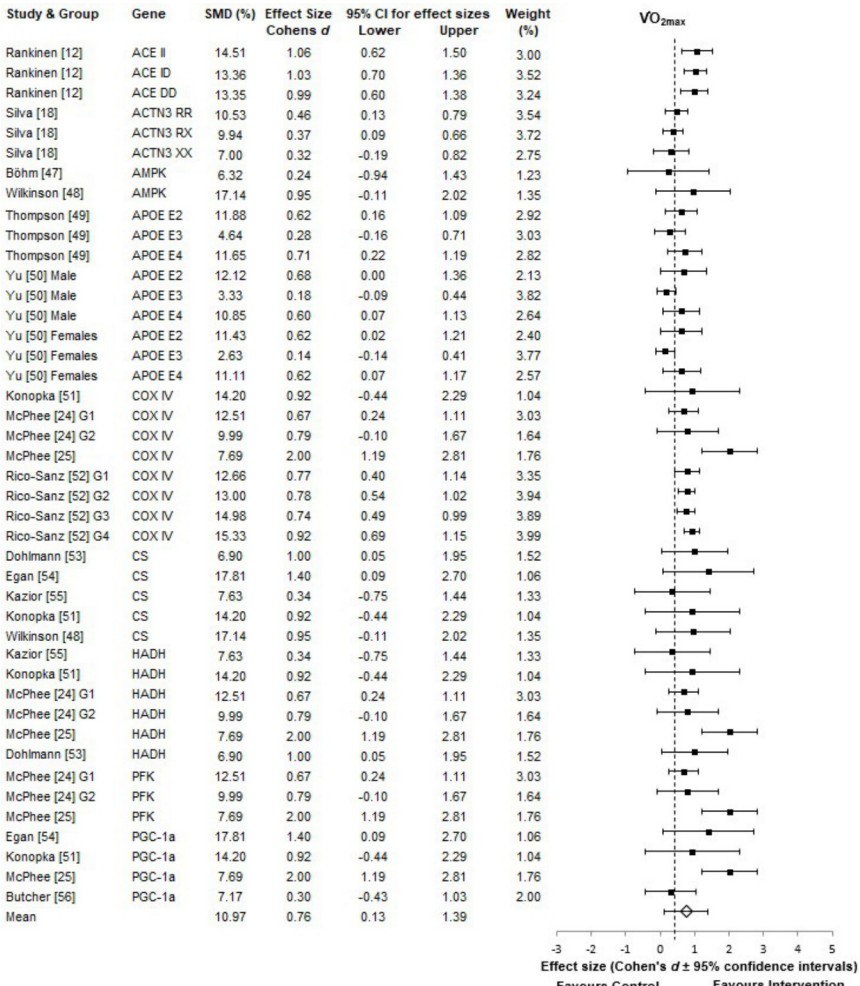

| Study & Group | Gene | SMD (%) | Effect Size Cohens d | 95% CI for effect sizes Lower | Upper | Weight (%) |
|---|---|---|---|---|---|---|
| Rankinen [12] | ACE II | 14.51 | 1.06 | 0.62 | 1.50 | 3.00 |
| Rankinen [12] | ACE ID | 13.36 | 1.03 | 0.70 | 1.36 | 3.52 |
| Rankinen [12] | ACE DD | 13.35 | 0.99 | 0.60 | 1.38 | 3.24 |
| Silva [18] | ACTN3 RR | 10.53 | 0.46 | 0.13 | 0.79 | 3.54 |
| Silva [18] | ACTN3 RX | 9.94 | 0.37 | 0.09 | 0.66 | 3.72 |
| Silva [18] | ACTN3 XX | 7.00 | 0.32 | -0.19 | 0.82 | 2.75 |
| Böhm [47] | AMPK | 6.32 | 0.24 | -0.94 | 1.43 | 1.23 |
| Wilkinson [48] | AMPK | 17.14 | 0.95 | -0.11 | 2.02 | 1.35 |
| Thompson [49] | APOE E2 | 11.88 | 0.62 | 0.16 | 1.09 | 2.92 |
| Thompson [49] | APOE E3 | 4.64 | 0.28 | -0.16 | 0.71 | 3.03 |
| Thompson [49] | APOE E4 | 11.65 | 0.71 | 0.22 | 1.19 | 2.82 |
| Yu [50] Male | APOE E2 | 12.12 | 0.68 | 0.00 | 1.36 | 2.13 |
| Yu [50] Male | APOE E3 | 3.33 | 0.18 | -0.09 | 0.44 | 3.82 |
| Yu [50] Male | APOE E4 | 10.85 | 0.60 | 0.07 | 1.13 | 2.64 |
| Yu [50] Females | APOE E2 | 11.43 | 0.62 | 0.02 | 1.21 | 2.40 |
| Yu [50] Females | APOE E3 | 2.63 | 0.14 | -0.14 | 0.41 | 3.77 |
| Yu [50] Females | APOE E4 | 11.11 | 0.62 | 0.07 | 1.17 | 2.57 |
| Konopka [51] | COX IV | 14.20 | 0.92 | -0.44 | 2.29 | 1.04 |
| McPhee [24] G1 | COX IV | 12.51 | 0.67 | 0.24 | 1.11 | 3.03 |
| McPhee [24] G2 | COX IV | 9.99 | 0.79 | -0.10 | 1.67 | 1.64 |
| McPhee [25] | COX IV | 7.69 | 2.00 | 1.19 | 2.81 | 1.76 |
| Rico-Sanz [52] G1 | COX IV | 12.66 | 0.77 | 0.40 | 1.14 | 3.35 |
| Rico-Sanz [52] G2 | COX IV | 13.00 | 0.78 | 0.54 | 1.02 | 3.94 |
| Rico-Sanz [52] G3 | COX IV | 14.98 | 0.74 | 0.49 | 0.99 | 3.89 |
| Rico-Sanz [52] G4 | COX IV | 15.33 | 0.92 | 0.69 | 1.15 | 3.99 |
| Dohlmann [53] | CS | 6.90 | 1.00 | 0.05 | 1.95 | 1.52 |
| Egan [54] | CS | 17.81 | 1.40 | 0.09 | 2.70 | 1.06 |
| Kazior [55] | CS | 7.63 | 0.34 | -0.75 | 1.44 | 1.33 |
| Konopka [51] | CS | 14.20 | 0.92 | -0.44 | 2.29 | 1.04 |
| Wilkinson [48] | CS | 17.14 | 0.95 | -0.11 | 2.02 | 1.35 |
| Kazior [55] | HADH | 7.63 | 0.34 | -0.75 | 1.44 | 1.33 |
| Konopka [51] | HADH | 14.20 | 0.92 | -0.44 | 2.29 | 1.04 |
| McPhee [24] G1 | HADH | 12.51 | 0.67 | 0.24 | 1.11 | 3.03 |
| McPhee [24] G2 | HADH | 9.99 | 0.79 | -0.10 | 1.67 | 1.64 |
| McPhee [25] | HADH | 7.69 | 2.00 | 1.19 | 2.81 | 1.76 |
| Dohlmann [53] | HADH | 6.90 | 1.00 | 0.05 | 1.95 | 1.52 |
| McPhee [24] G1 | PFK | 12.51 | 0.67 | 0.24 | 1.11 | 3.03 |
| McPhee [24] G2 | PFK | 9.99 | 0.79 | -0.10 | 1.67 | 1.64 |
| McPhee [25] | PFK | 7.69 | 2.00 | 1.19 | 2.81 | 1.76 |
| Egan [54] | PGC-1a | 17.81 | 1.40 | 0.09 | 2.70 | 1.06 |
| Konopka [51] | PGC-1a | 14.20 | 0.92 | -0.44 | 2.29 | 1.04 |
| McPhee [25] | PGC-1a | 7.69 | 2.00 | 1.19 | 2.81 | 1.76 |
| Butcher [56] | PGC-1a | 7.17 | 0.30 | -0.43 | 1.03 | 2.00 |
| Mean | | 10.97 | 0.76 | 0.13 | 1.39 | |

**Fig 2. $\dot{V}O_{2max}$ forest plot.** Effect sizes represent the change in $\dot{V}O_{2max}$ post-intervention. For all plots the 95% confidence intervals were calculated and the overall mean effect size is represented with the diamond, whereas, the black squares are the individual effect size of each study. The weighting is adjusted for sample size, SD and variance. Where G1 = group 1 and G2 = group 2. Genes listed in alphabetical order.

**Table 1. Candidate genes for cardio-respiratory fitness.**

| Gene Groups | No. of Study groups | Total Group Sample Size | Mean Rank (Not adjusted for sample size) | Subgroup (Adjusted for sample size) % of weight | Mean Group Effect Size (d) |
|---|---|---|---|---|---|
| ACE | 3 | 188 | 35.33 | 14.24 | 1.02 |
| ACTN3 | 3 | 206 | 8.33 | 14.15 | 0.39 |
| AMPK/PRKAA2 | 2 | 18 | 17.25 | 7.84 | 0.62 |
| APOE | 9 | 437 | 10.44 | 13.67 | 0.41 |
| COX4I1 | 8 | 591 | 25.06 | 13.43 | 0.85 |
| CS | 5 | 46 | 28.00 | 12.12 | 0.90 |
| HADH | 6 | 106 | 25.25 | 10.58 | 0.96 |
| PFKM | 3 | 78 | 27.17 | 6.62 | 1.12 |
| PGC-1α | 4 | 52 | 28.25 | 7.36 | 1.14 |
| Total | 43 | 1,722 | | | |

The group with the highest mean rank shows the greatest number of high scores in $\dot{V}O_{2max}$ and subgroup % shows how much of the overall 44% variance is accounted for by each of the nine genes, adjusted for sample size, SE and within group variation in SMD.

Within this subgroup the ACE gene, regardless of allele, had the greatest mean rank and weight on the increase in $\dot{V}O_{2max}$. ACE alleles (II, ID, DD) showed no significant difference in effect sizes. Other "aerobic" genes (COX4I1, CS, HADH, PFKM and PGC-1α) showed no significant differences between mean rank scores and contributed equally to the increase in $\dot{V}O_{2max}$. However, these genes, which all showed a large effect size, had a higher mean rank than those of ACTN3, AMPK and APOE groups. Interestingly, for the APOE genotype, which showed medium effect sizes for certain alleles, the E3/E3 combination, in both males and females, showed the lowest effect size on $\dot{V}O_{2max}$, at 0.18 and 0.14, respectively (Fig 2).

Here an F test found that there were no significant differences in APOE alleles effect size scores across genders for E2, E3 and E4; $p = 0.667$, $0.488$ and $0.776$, respectively. However, further analysis revealed that, as a whole, there were highly significant differences between allele groups, regardless of gender $F(2,6) = 59.52$, $p = 0.000$. Paired samples t-test found significant differences when comparing E2/E3 allele scores with E3/E3 at $p = 0.013$, similarly when comparing E3/E4 with E3/E3, $p = 0.002$. There was, however, no significant difference when comparing E2 and E4 ($p = 0.952$). Post-hoc LSD test found that E3 allele was the least effective, compared to both E2 and E4, in terms of its effect size.

## Genes associated with muscle strength (1RM)

Strength training intervention studies [11,23,48,52,55,57–61] found an average increase in lower body 1RM of 22.12 ± 10.08% across the study groups, with an overall large effect size, which was very highly significant (p < 0.001) (Fig 3). Typically these studies did not report activity duration, rather number of repetitions performed, which on average was 174 reps, session at intensities of 75% 1RM, over 3-days per-week and 10 weeks of training.

Again, 1RM did not meet the requirements for parametric testing between the six strength gene subgroups when split, D(29), 0.886, p < 0.05. Kruskal-Wallis H test found very highly significant difference between the gene specific groups; H (5), 20.081, p = 0.001. Partial Eta Squared tests, revealed that 72% of the total variability in the increase of 1RM strength post-training intervention, was explained by the genetic subgrouping (Table 2).

All strength associated gene groups, identified, were found to have a large influence in the variability of lower body 1RM, with large effect-sizes, except for the ACTN3 group, which had

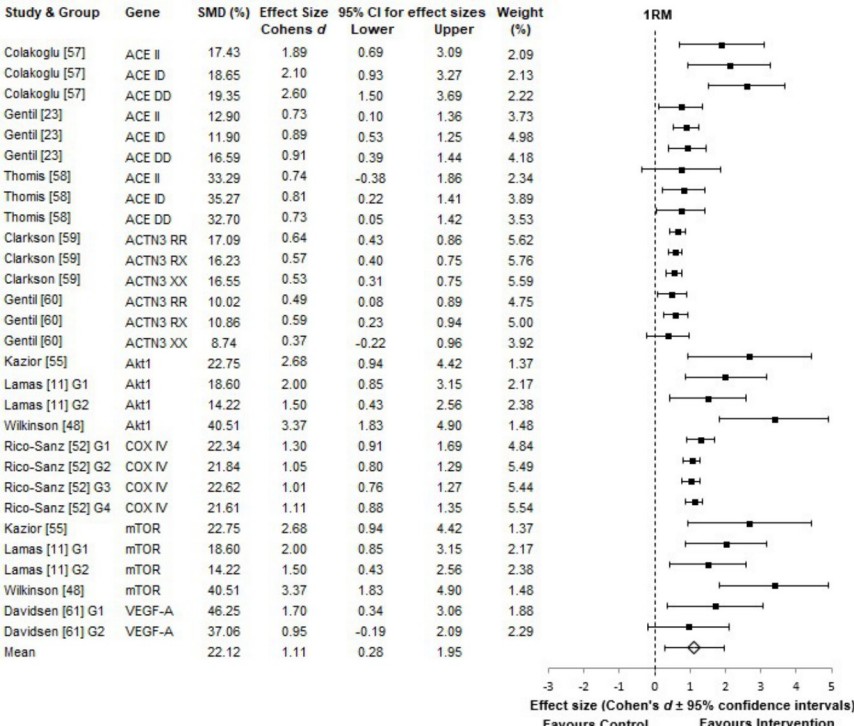

**Fig 3. 1RM forest plot.** The effect sizes are the change in 1RM post-intervention. For all plots the 95% confidence intervals were calculated and the overall mean effect size is represented with the diamond. The black squares represent the effect size of each study. Weighting was adjusted for sample size, SD and variance. Genes listed in alphabetical order.

a medium effect. This relative lack of effect was noted, regardless of ACTN3 genotype and allele (RR, RX, XX).

## Genes associated with anaerobic power

Here the analysis revealed a mean increase in peak power output of 12.17 ± 4.40%, irrespective of gene groups. Forest plot analysis found all studies [51,53,62–64] increased PPO by either a medium, or large effect-size (smallest ES 0.57), with a very highly significant improvement

**Table 2. Candidate genes for strength.**

| Gene Groups | No. of Study groups | Total Group Sample Size | Mean Rank (Not adjusted for sample size) | Subgroup (Adjusted for sample size) % of weight | Mean Group Effect Size (d) |
|---|---|---|---|---|---|
| ACE | 9 | 194 | 14.11 | 18.22 | 1.12 |
| ACTN3 | 6 | 743 | 3.50 | 22.96 | 0.57 |
| AKT1 | 4 | 39 | 24.00 | 11.71 | 2.27 |
| COX4I1 | 4 | 506 | 15.50 | 22.72 | 1.09 |
| mTOR | 4 | 39 | 24.00 | 11.71 | 2.27 |
| VEGF-A | 2 | 17 | 16.50 | 12.68 | 1.27 |
| Total | 29 | 1,538 | | | |

The group with the highest mean rank shows the greatest number of high scores in 1RM. Subgroup % shows how much each of the six genes contribute to the 72% variance, adjusted for sample size, SE and within group variation in SMD.

with training interventions (p < 0.001) (Fig 4). These studies performed between 4–12 cycle bouts, at average intensities of 90–110% $\dot{V}O_{2max}$ or load of 0.075 per-kg bodyweight, over 3-days a week and 5 weeks of training.

As with previous data sets, PPO results were not normally distributed, D(17), .888, p < 0.05. However, there were no significant differences between gene subgroups, H(3), 1.592, p = 0.661 and partial Eta Squared found that only 10% of the variability in the 12.17% increase in PPO post-training, was explained by genetic sub-grouping (Table 3).

All genes showed a medium, or large effect size, with the HADH gene group showing the largest effect size score, (d = 1.34). However, HADH showed the lowest weighting, of 18.95%, when compared to other genes. MAFbx, with an effect size of 0.70, carried 34.36% of the weighting and explained about 1/3 of the total variability in the 10% power increase.

## Discussion

The aim of this systematic review and meta-analysis was to identify candidate genes associated with the three key components of fitness. Additionally, to assess if these genes and their alleles, are associated with exercise response phenotype variability, in untrained human subjects, following an exercise training intervention. The results from this review are important, not just for the untrained, but all training populations. The inter-individual variation in the improvements in health-related components of fitness, identified in this study, for $\dot{V}O_{2max}$, strength and power can be explained genetics up to 44, 72 and 10% respectively. Such a finding emphasises the importance of assessing individuals' genotype and planning training accordingly, thereby, making these findings relevant to the wider field of sport and exercise sciences.

13 candidate genes were identified that show significant associations with the fitness variables of interest. Overall, this review found that a genetic component for exercise responsiveness can explain between 10–72% of the variability in these key components. Such findings are consistent with previous studies, which reported variabilities of up to 80% in fitness phenotypes [2,13,15,17,27]. In this review, the subgroup analysis of the 13 candidate genes showed that nine were associated with cardiovascular fitness, six with muscular strength and four with anaerobic power phenotypes. Although studies, reported the ACE, ACTN3 and APOE allelic

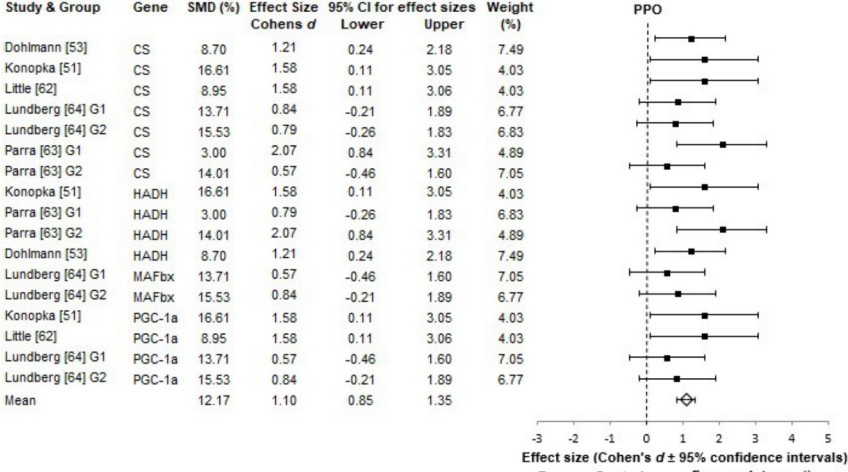

**Fig 4. PPO forest plot.** ES represents the change in PPO post-intervention. For all plots the 95% CI's were calculated and the overall mean effect size is represented with the diamond, whereas the black squares are the effect size of each study. The weight is adjusted for sample size, SD and variance. Genes listed in alphabetical order.

Table 3. Candidate genes for peak power output.

| Gene Groups | No. of Study groups | Total Group Sample Size | Mean Rank (Not adjusted for sample size) | Subgroup (Adjusted for sample size) % of weight | Mean Group Effect Size (d) |
|---|---|---|---|---|---|
| CS | 6 | 66 | 9.33 | 26.15 | 1.15 |
| HADH | 4 | 39 | 10.88 | 18.96 | 1.34 |
| MAFbx | 3 | 27 | 6.17 | 34.36 | 0.70 |
| PGC-1a | 4 | 34 | 8.75 | 20.53 | 1.02 |
| Total | 17 | 166 | | | |

The group with the highest mean rank shows the greatest number of high scores in PPO and subgroups show how much of the 10% variance is accounted for by each of the four genes when normalised to 100%, adjusted for study sample size, SE and within group SD in SMD.

contribution, a key limitation with the majority of studies, was that they assessed individual genes, as a single variable, irrespective the gene's allelic composition. Such an omission makes it difficult to assess the exact role of the gene(s) as the minor and major alleles often affect the phenotype differently, as shown in this study with APOE alleles [17,18]. Different genes also interact to produce the final phenotypic response [65], and here Genome Wide Association Studies (GWAS) will play an increasingly important role in identifying these and the variants. Thus, Williams et al. [65] identified a total of 97 genes that predicted $\dot{V}O_{2max}$ trainability, and that the phenotype was dependent on several of these genotypes, which may contribute to approximately 50% of an individual's $\dot{V}O_{2max}$ trainability [65]. Further, a recent study by Al-Khelaifi et al. [66] also uncovered novel genes and associations using GWAS [66]. Using these studies that have identified the genes and potential associations to training this study identified 13 candidate genes, that provides a useful focus for future exercise intervention studies and how the variability of the phenotypes are affected.

In terms of cardiorespiratory fitness, all studies demonstrated an increased $\dot{V}O_{2max}$ in response to aerobic exercise interventions. Here the well-studied ACE gene and its polymorphisms; II, ID, DD, showed the greatest influence on the phenotype, despite only having three groups in the analysis, with little differences between genotypes (188 participants out of 1,772, Table 1). Following this, COX4I1, CS and HADH genes also showed large influences on aerobic improvements. A possible explanation is that these genes code for key mitochondrial enzymes used in aerobic respiration. Thus, COX4I1 codes for cytochrome C oxidase, a key component of mitochondrial electron transport, whilst CS codes for citrate synthase, found in Krebs' Cycle. Finally, HADH codes for 3-hydroxyacyl-CoA dehydrogenase, required for the oxidation of fatty acids [15,17,25,67,68]. PFKM and PGC1-α also displayed large influences and effect sizes, but only contributed 6.62 and 7.36% of the total 44% gene variability, respectively. A possible explanation is the combination of low study numbers and sample sizes (78 and 52 out of the 1,722 participants). AMPK had the smallest sample size, of 18 participants, hence it is difficult to draw firm conclusions on its effect on cardiorespiratory fitness, when compared to the other genes. However, AMPK has been found to directly affect PGC1-α, which is the independent master regulator of mitochondrial biogenesis and aerobic respiration [68,69]. Finally, APOE and ACTN3 results indicated no advantages and little influence on cardiorespiratory fitness. The APOE E3/E3 allele is very common, at 78.3%, worldwide [70] and is considered the neutral genotype, showing no effect on cardiorespiratory fitness, in agreement with the findings in this review. However, high to medium effect sizes were observed for the E2 and E4 alleles (Fig 2), concurring with the findings of Bernstein et al. [71] and Deeny et al. [72]. In this connection Obisesan et al. [73] found APOE genotypes explained significant variability in cardiorespiratory fitness, seen in training-induced increases after 24

weeks (p = 0.002). Further analysis in this review found that there was a highly significant difference when the E3 allele was compared to E2 and E4, in both males and females. This could explain the observed 44% genetic influence seen in this review, and why it was lower than reported rates in the literature, due to the confounding effect of APOE E3/E3 alleles [2,15,16,74,75]. Additionally, this would also emphasise the importance of examining the contributions of specific alleles of candidate genes, as opposed to whole gene analysis. These findings would also suggest that it is more advantageous to carry E2 and E4 genotypes, as opposed to E3, for improvements of $\dot{V}O_{2max}$ post-training. Raichlen and Alexander, [70] stated, that these specific candidate genes and genotypes do not necessarily aid physical performance phenotypes, such as cardiorespiratory fitness, but instead, in the presence of physical activity, decrease health risks, such as coronary artery disease (CAD) and improve overall health status, therefore, when this is included in the meta-analysis it consequently decreases the associated gene variability [76].

Interestingly, the well-researched ACTN3 gene showed equivocal results in this review for the both $\dot{V}O_{2max}$ and 1RM. Theoretically, homozygosity for the X (deletion) allele should abolish production of α actinin-3, leading to improved aerobic fitness, whilst the R allele should decrease aerobic fitness, due to increased α actinin-3 expression [18,27,60,61,77–79]. Additionally, Hogarth et al. [80] states that α-actinin-3 controls sarcomeric composition and muscle function in an allele dose-dependent fashion and promotes strength adaptations, but this was not observed in this review. However, the findings of this review agree with those of Gineviciene et al. [1], which directly assessed the influence of the ACTN3 R577X polymorphism on aerobic fitness and found no significant differences between alleles for the variability and trainability of the exercise phenotype [14]. Similarly, for the well-studied ACE genotypes, this review found no significant differences between ACE insertion (I) and deletion (D) alleles, as both alleles were associated with significant improvements in cardiorespiratory fitness. Here previous work has suggested that I allele is associated with greater increases in cardiorespiratory fitness and endurance, due to lower levels of ACE and increased maximal heart rate and improved blood circulatory role [58].

In this review, muscular strength phenotype, assessed by 1RM, showed the largest observed variability, in response to training interventions, with 72% accounted for by the six candidate genes. AKT1 and mTOR had equally large contributions to the phenotype response, displaying the largest mean rank and effect size. Previous studies [81–83] are consistent in showing interactions between Akt and mTOR regulation, which are activated through resistance exercise. Akt and its downstream signalling pathways, such as mTOR (Akt-mTOR pathway) is the central mediator of protein synthesis, associated with the control of skeletal muscle hypertrophy, muscle mass and strength [80–83]. However, due to the nature of the studies, reporting the findings from whole gene analysis, it is still unclear on the specific role of different alleles. Interestingly, the literature supports an upstream regulation in AMPK activation for endurance, suppressing Akt-mTOR, meaning the increased levels of AMPK may be detrimental to strength improvements [84–87]. Additionally, the literature suggests that mTOR polymorphism (rs2295080) alleles G = 45.2% and T = 54.8% also show different results [88]. The G-allele predominates in endurance athletes, whilst the T-allele frequency is greater in power and strength-oriented athletes [89]. Therefore, suggests the need to review this gene at an allele specific level.

ACE and VEGF-A made similar contributions to increased strength variability, despite the low number of participants (17 participants) in the VEGF-A study. The results further show that ACE and COX4I1 accounted for 40.9% of the 72% total variability found. Theoretically the ACE D allele results in increased ACE activity, which has been shown to be associated with

strength performance and decreases in $\dot{V}O_{2max}$ [23,29,90], however no such effect was noted in this study. Finally, ACTN3 had the lowest mean rank score, and the lowest effect size, when compared to the other genes. However, it is important to note that the increases in strength, associated with ACTN3, were still significant, which is consistent with previous findings [27,59,91]. Moreover, when sub-group analysis and sample size were accounted for, ACTN3 was the largest contributor to overall variability in the strength phenotype. However, this may simply reflect the high proportion of participants within this group (743 out of the 1,538).

PPO displayed the lowest genetic influence (10% variability) but was still significantly improved by the training intervention. Here one new candidate gene was identified, by this review, that had not already been linked with another component of fitness, the MAFbx (FBXO32), or Atrogin-1 gene. Despite low study numbers (27 participants), MAFbx accounted for 34.36% of the total 10% variability. MAFbx has previously been found to be associated with muscle strength gains during exercise induced muscle hypertrophy [65,92–94]. In agreement, Mascher et al. [93] found that this gene is involved in muscle breakdown and atrophy, and that resistance training reduced its expression, hence on this basis, might be associated with strength. Such findings reflect a paucity of investigations into candidate genes for the PPO phenotype, as only five studies were included in this review. Moreover, the current analysis also revealed three of the identified genes, were also associated with cardiorespiratory fitness. Again, the allele compositions of these genes might have been more informative but were not reported. Previous work [1] has identified that PPARGC1A gene (PGC-1α) is associated with power variables, as has this review.

It is important to note that, one possible reason why the anaerobic power candidate genes may be associated with different phenotype responses and low variability rates, could be due to studies incorrectly measuring anaerobic power and rather, measuring metabolic power, which is a mixture of energy sources. This may be a significant flaw in many power assessment studies [10]. In this connection, it is very common, when measuring power, to use 30 second Wingate tests (WAnT). Here energy from anaerobic phosphagenic, glycolytic and aerobic mitochondrial respiration metabolism, contributes to 31.1%, 50.3% and 18.6%, respectively [8,95]. Hence, many studies are misinterpreting power phenotypes, making the assessment more difficult. Therefore, we would recommend studies that include, all-out burst and brief sprints with durations of up to 10 seconds, where the initial energy source is primarily drawn from anaerobic metabolism only, and reported as peak power output, rather than mean power output over time [10,96,97].

A major strength of this meta-analytical was the ability to compare all studies, regardless of intervention, by grouping studies and assessing them by the genotype sub-groups. For example, in this review, all studies that assessed the same genes and alleles were combined, any effect on the phenotype was averaged and the overall variability assessed. This was then compared with the influences of the other candidate genes following the same method, rather than directly comparing studies against each other using different interventions. Such an approach helps account for the variation caused by the training intervention and other external influences, such as the environment on the phenotype. Another key strength of this review was that the analysis compared the contribution of multiple genes towards the total variance of the phenotypes, rather than a more restricted, single gene analysis approach. It is also important to note that the genetic make-up, alone, does not determine the phenotype, only the potential for expression of the phenotype in response to a particular lifestyle, environment, and intervention [13,17].

The main limitation to this review was the lack of allele specific analysis in the included studies. Another limitation is the possible exclusion of other candidate genes, potentially

reducing the number of candidate genes identified. Such limitations reflect a generic problem with a systematic review, that it is limited to current published information and the requirement to ensure the comparability of different studies. Another factor to consider, is that baseline-training status also affects the adaptation responses to exercise. Although, this review attempted to address this by specifically selecting studies in which all participants were classed as untrained, according to widely accepted norms, it is clear there were still baseline differences between individuals. This is reflected in this review by the within-groups non-normal distributions and heterogeneity [2,4]. Moreover, the predisposition of the genetic heritability for advantageous baseline phenotypes, shows genes and specific alleles heavily influence adaptations and trainability, even before training interventions are implemented [14]. Finally, for a number of particular genes in this review, due to the low group sample size, it is not clear, nor possible to draw firm conclusions for the precise role of these genes on the phenotype and further investigation is required. Nevertheless, this review has identified 13 candidate genes, which explain a significant proportion of the variability and their contribution in the phenotype responses to trainability for the three components of fitness.

## Practical applications

This review demonstrates that the candidate genes provide valuable information regarding genotype-specific training and variability in the phenotype responses. In theory a possible practical application of this could be to identify a person's genotype and tailor a specific individual training intervention programme, based on their genotype. This would be more advantageous than implementing generic training programmes, which may provide relatively little value in terms of phenotype gains and improvements. These inferences also support and strengthen the evidence, that genes have on training variability suggested in the research literature.

## Supporting information

**S1 Fig. Bland Altman plot.** The results of both reviewers using the quality assessment tool is mapped as the difference in scores against the average score (Bias). The 95% LoA are also calculated and represented as the upper and lower 1.96 dashed lines. The confidence intervals for the 95% LoA were calculated using Bland Altman's approximate method.
(PDF)

**S1 Table. Search terms and results.** Search terms implemented for all the databases and the number of results shown by hits.
(PDF)

**S2 Table. COSMIN assessment tool and Post-Hoc power.** The average score between reviewers was taken for the final inclusion. Power threshold was based at 0.8 (80%).
(PDF)

**S1 File. Meta-analysis on genetic association studies checklist.**
(PDF)

**S2 File. PRISMA 2009 checklist.**
(PDF)

## Acknowledgments

This meta-analytical review is non-profited and non-funded research. This work was supported by Anglia Ruskin University, the staff from library services and the Faculty of Science

and Engineering (FSE). All data is available at the Cambridge Centre for Sport & Exercise Sciences, Anglia Ruskin University, UK.

## Author Contributions

**Conceptualization:** Henry C. Chung, Don R. Keiller, Justin D. Roberts, Dan A. Gordon.

**Data curation:** Henry C. Chung, Don R. Keiller.

**Formal analysis:** Henry C. Chung.

**Investigation:** Henry C. Chung, Don R. Keiller.

**Methodology:** Henry C. Chung, Dan A. Gordon.

**Project administration:** Henry C. Chung.

**Resources:** Don R. Keiller, Justin D. Roberts, Dan A. Gordon.

**Software:** Dan A. Gordon.

**Supervision:** Don R. Keiller, Justin D. Roberts, Dan A. Gordon.

**Validation:** Henry C. Chung, Dan A. Gordon.

**Visualization:** Henry C. Chung, Don R. Keiller, Justin D. Roberts, Dan A. Gordon.

**Writing – original draft:** Henry C. Chung.

**Writing – review & editing:** Henry C. Chung, Don R. Keiller, Justin D. Roberts, Dan A. Gordon.

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
