## [Decision Letter · Decision Letter 0]

24 Jun 2021

PONE-D-21-07079

Do exercise-associated genes explain phenotypic variance in the three components of fitness? A Systematic review & Meta-analysis.

PLOS ONE

Dear Dr. Chung,

Thank you for submitting your manuscript to PLOS ONE. After careful consideration by two expert reviewers, we feel that it has merit but does not fully meet PLOS ONE’s publication criteria as it currently stands. Therefore, we invite you to submit a revised version of the manuscript that addresses the points raised during the review process.

Please address each point that has been raised by the expert reviewers and revise your manuscript accordingly. In the revision please also consider discussing a  genome wide analysis in response to training interventions in your study.

We look forward to receiving your revised manuscript.

Kind regards,

Stephen E Alway, Ph.D.

Academic Editor

PLOS ONE

Journal Requirements:

Reviewers' comments:

Reviewer's Responses to Questions

**Comments to the Author**

1. Is the manuscript technically sound, and do the data support the conclusions?

Reviewer #1: Yes

Reviewer #2: Yes

2. Has the statistical analysis been performed appropriately and rigorously? 

Reviewer #1: Yes

Reviewer #2: Yes

3. Have the authors made all data underlying the findings in their manuscript fully available?

Reviewer #1: Yes

Reviewer #2: Yes

4. Is the manuscript presented in an intelligible fashion and written in standard English?

Reviewer #1: Yes

Reviewer #2: Yes

5. Review Comments to the Author

Reviewer #1: PONE-D-21-07079

Do exercise-associated genes explain phenotypic variance in the three components of fitness? A Systematic review & Meta-analysis.

General comments: Regular physical activity is important to promote health benefits and exercise tolerance in those who are inactive. The Authors have performed a thorough review examining potential genes related to the training response. This is interesting work and of interest to the Journal’s audience.

Specific comments: Please respond to the comments listed below regarding your work; thank you.

Abstract is well written yet I would like to see you denote the gender breakdown of participants as well as their mean age. Also, since this is in untrained participants, is this really that important considering that these individuals are not using these outcomes on a day-to-day basis? Does the merit of this then solely relate to training studies performed in this sample trying to better understand how these factors changes in response to various training regimens?

Introduction; Lines 52-56 should also denote that VO2max is a correlate or morbidity and mortality according to Blair et al. (1996), etc.

Line 57: isn’t strength the maximal force generated by muscle?

Line 73: but my recollection of these data which is a little foggy is that the relationships (R2 value) between these genes and measures such as VO2max are not that strong e.g. < 20 %, so clearly other attributes have a stronger association with this measure than a gene or two.

Line 93: is a hypothesis merited here?

Methods; line 117: is non-obese also an inclusion criterion? Please describe. And were studies only included if they did not have any nutritional intervention such as protein intake, etc. to optimize changes in strength?

Line 120: is this treadmill derived VO2max or that on the cycle ergometer? And I assume that this is leg press derived measures of 1RM and Wingate test derived indices of Anaerobic power? For example, load determines PPO from the Wingate test so if a study used a relatively light training load, then resultant PPO is lower. Is this a limitation of your criteria?

Study selection: were studies included and excluded using some software, or did 2 or 3 of the authors meet and talk about whether a study was included or not? Please explain this in the text and how any disagreements were handled; thank you.

Results, line 178: can you please add to your text here the range of training duration of these studies eliciting this near 11 % increase in CRF; thank you. And please do the same for the change in strength and AN power.

Table 1: can you please explain here in the legend what ‘mean rank’ refers to, as you did with the other outcome = subgroup?

Discussion, line 267: Please finish this section of text here with a brief argument or rationale as to why this review and its findings matter to the broader field of Sports Science.

Line 272: I do not do this genetic work, so this may be a dumb question, but what does it mean when a single gene is related to training responsiveness for each of these 3 outcomes which are physiologically limited by different factors i.e. O2 delivery for CRF, muscle mass, FT fiber %, and neuromuscular function for 1RM and PPO?

Does gene expression change based on the training volume or duration performed in these studies? If so, how does this affect your resultant data?

Line 322: ‘equivocal results’ for what outcome or factor?

Line 353: I believe this link needs to be replaced with a reference citation.

Reviewer #2: The aim of this manuscript was to report candidate genes associated with three components of exercise (i.e., cardiovascular fitness, strength, and aerobic power) through systematic review and meta-analysis. The authors identified 24 manuscripts meeting all inclusion criteria, whereby an exercise training intervention was undertaken in untrained individuals, with pre-post phenotypic measures assessed. Importantly, the investigators identified 13 candidate genes associated with phenotypic responses to exercise, with some degree of variability in exercise responses potentially explained by genetic influences.

The importance of key genes responsible for the modulation of responses to various forms of exercise training are well known, with limited data regarding the collective influence of multiple genes. The manuscript is written in a logical manner, and the analysis undertaken suitable. While the authors do address the role of each candidate gene, further discussion into the context of their role in regard to previous publications should be made (e.g., gene knockout/ in studies). Moreover, as genome wide analysis in response to training interventions are becoming more frequent, the authors should discuss their findings in context of these.

Additional comments:

Line 364 – This line is potentially misleading. While testosterone plays a role, AKT/mTOR are the key regulators of muscle mass/strength.

Lines 351-357 – The reason for these lines are unclear, while mTOR polymorphisms may have a role in response to endurance vs strength bases training, these were not assessed in the studies analysed.

Lines 412- 414 – The first limitation is unclear

Line 436-437 – Could the authors clarify this sentence; it is currently speculative to tailor training interventions based upon an individual’s genotype.

Table S2 – Could the authors provide numbered references against each of the studies.

6. PLOS authors have the option to publish the peer review history of their article (what does this mean?). If published, this will include your full peer review and any attached files.

Reviewer #1: No

Reviewer #2: No

---

## [Author Response · Author response to Decision Letter 0]

4 Aug 2021

All specific responses to comments are found in the attached document "Response to reviewers". 

We beleive we have responsed and addressed all comments outlined within the perviouis manuscript and that this ammended version meets the PLOS ONE requirements and hope it is satificatiory for the reviewers / editors.

---

## [Decision Letter · Decision Letter 1]

4 Oct 2021

Do exercise-associated genes explain phenotypic variance in the three components of fitness? A Systematic review & Meta-analysis.

PONE-D-21-07079R1

Dear Dr. Chung,

We’re pleased to inform you that your manuscript has been judged scientifically suitable for publication and will be formally accepted for publication once it meets all outstanding technical requirements.

Kind regards,

Stephen E Alway, Ph.D.

Academic Editor

PLOS ONE

Additional Editor Comments (optional):

Reviewers' comments:

Reviewer's Responses to Questions

**Comments to the Author**

1. If the authors have adequately addressed your comments raised in a previous round of review and you feel that this manuscript is now acceptable for publication, you may indicate that here to bypass the “Comments to the Author” section, enter your conflict of interest statement in the “Confidential to Editor” section, and submit your "Accept" recommendation.

Reviewer #1: All comments have been addressed

Reviewer #2: All comments have been addressed

2. Is the manuscript technically sound, and do the data support the conclusions?

Reviewer #1: Yes

Reviewer #2: Yes

3. Has the statistical analysis been performed appropriately and rigorously? 

Reviewer #1: Yes

Reviewer #2: Yes

4. Have the authors made all data underlying the findings in their manuscript fully available?

Reviewer #1: Yes

Reviewer #2: Yes

5. Is the manuscript presented in an intelligible fashion and written in standard English?

Reviewer #1: Yes

Reviewer #2: Yes

6. Review Comments to the Author

Reviewer #1: I appreciate the care taken by the Authors to respond fully to my initial points raised and developing a very extensive rebuttal and changes to their paper. This is a much improved work that should be of interest to those who do work in this area.

Reviewer #2: All my previous comments and concerns have been addressed and the manuscript is greatly improved.

7. PLOS authors have the option to publish the peer review history of their article (what does this mean?). If published, this will include your full peer review and any attached files.

Reviewer #1: No

Reviewer #2: No

---

## [Editor Report · Acceptance letter]

6 Oct 2021

PONE-D-21-07079R1 

Do exercise-associated genes explain phenotypic variance in the three components of fitness? A Systematic review & Meta-analysis. 

Dear Dr. Chung:

I'm pleased to inform you that your manuscript has been deemed suitable for publication in PLOS ONE. Congratulations! Your manuscript is now with our production department. 

Kind regards, 

on behalf of

Dr. Stephen E Alway 

Academic Editor

PLOS ONE